# Cytomegalovirus Immunity, Inflammation and Cognitive Abilities in the Elderly

**DOI:** 10.3390/v13112321

**Published:** 2021-11-21

**Authors:** Jacqueline Hesson, Neva Fudge, Michael Grant

**Affiliations:** 1Faculty of Education, Memorial University of Newfoundland, St. John’s, NL A1B 3X8, Canada; jhesson@mun.ca; 2Immunology and Infectious Diseases Program, Division of BioMedical Sciences, Faculty of Medicine, Memorial University of Newfoundland, St. John’s, NL A1B 3V6, Canada; nfudge@mun.ca

**Keywords:** cytomegalovirus, immunity, inflammation, aging, cognitive decline, fractalkine

## Abstract

Reducing the socioeconomic toll from age-related physical and mental morbidities requires better understanding of factors affecting healthy aging. While many environmental, lifestyle, and genetic factors affect healthy aging, this study addressed the influence of cytomegalovirus (CMV) infection and immunity on age-related inflammation and cognitive abilities. Healthy adults 70–90 years old were recruited into a prospective study investigating relationships between anti-CMV immunity, markers of inflammation, baseline measures of cognitive ability, and changes in cognitive ability over 18 months. Humoral and cellular responses against CMV, levels of inflammatory markers, and cognitive abilities were measured at study entry, with measurement of cognitive abilities repeated 18 months later. CMV-seropositive and -seronegative sub-groups were compared, and relationships between anti-CMV immunity, markers of inflammation, and cognitive ability were assessed. Twenty-eight of 39 participants were CMV-seropositive, and two had CMV-specific CD8^+^ T cell responses indicative of CMV immune memory inflation. No significant differences for markers of inflammation or measures of cognitive ability were observed between groups, and cognitive scores changed little over 18 months. Significant correlations between markers of inflammation and cognitive scores with interconnection between anti-CMV antibody levels, fractalkine, cognitive ability, and depression scores suggest areas of focus for future studies.

## 1. Introduction

Healthier aging is an increasingly urgent socioeconomic objective, fueled in large part by the generalized demographic shift towards more elderly populations [1]. Age-related morbidities such as Alzheimer’s disease, cancer, cardiovascular disease, and type II diabetes impose a massive multifaceted burden on individuals, families, the healthcare system, and society [2]. Mental health issues such as depression, anxiety, and cognitive decline present further challenges to healthy aging [3]. While some physiological and cognitive decline with aging may be unavoidable, numerous environmental factors and interactions influence the pace of decline. There is considerable evidence that a worsening background of low-grade chronic inflammation (termed inflammaging) elevates the risk for developing age-related physical and psychological morbidities [4,5]. In this context, accumulation of senescent cells is linked to age-related deterioration of health [6,7]. Senescent cells arise following extensive cell division or exposure to radiation, oxidative stress, toxins, or other environmental insults. Affected cells acquire a cell type-specific senescence associated secretory phenotype (SASP), often characterized by release of proinflammatory cytokines such as interleukin-6 (IL-6) [8]. Aging of the immune system itself may trigger or promote inflammaging, both by contributing senescent immune cells and, as its capacity for effective surveillance falls, by being more tolerant of their persistence. Thus, unresolved inflammation can initiate and propagate a vicious cycle of immune system aging and immune cell senescence leading to more inflammation.

One common environmental factor that is recognized to accelerate aging of the immune system is infection with cytomegalovirus (CMV), a ubiquitous human β-herpes virus [9]. While CMV inevitably establishes chronic infection, it remains predominantly latent and asymptomatic in healthy individuals [10]. However, over a lifetime of infection, a significant fraction of CMV-infected elderly individuals experience abnormal expansions of CD8^+^ T cells reactive against CMV, which manifest in a characteristic immune phenotype associated with the immune risk profile (IRP) [11]. The characteristic phenotype includes increased frequencies of terminally differentiated (CD57^+^) T cells, inverted CD4^+^/CD8^+^ T cell ratios (<1.0), and reduced B cell numbers [12]. The IRP is functionally reflected in lesser mitogen-induced T cell proliferation in vitro, poor in vivo vaccine responsiveness, and accelerated all-cause mortality [13]. This age-related phenomenon is uniquely associated with CMV infection, prompting a range of studies exploring the relationship between CMV infection and healthy aging [14,15,16,17,18,19].

Despite the clear relationship between CMV infection and immune system aging, no consensus has emerged as to CMV infection being associated with greater inflammaging or increasing the risk for earlier or more prominent development of age-related morbidities. As the IRP is defined by features related to the immune response against CMV, we questioned whether certain attributes of anti-CMV immunity itself, rather than acute responses to CMV reactivation, might be associated with inflammaging or with an increased or accelerated risk for age related morbidities. Higher levels of anti-CMV antibodies were associated with impaired health and reduced cognitive function in very old subjects and with reduced cognitive function in non-elderly adults [20,21]. Mechanistic links between anti-CMV immunity and cognitive decline remain speculative, but pro-inflammatory anti-CMV effector responses that accumulate through life and persist through periods of viral latency may play a role as reduced cognition and subsequent depression have both previously been tied to inflammation [22].

In this study of individuals over 70 years old, we measured immune responses against CMV and a number of plasma markers of inflammation at study entry, then administered a set of standardized neuropsychological tests on two occasions 18 months apart to assess cognitive status at entry and changes in cognitive status over an 18 month period. The aim was to explore possible links between CMV infection, anti-CMV immunity, inflammation, and cognitive decline in an elderly population. Our general hypothesis was that inflationary anti-CMV immunity would be associated with increased inflammation and cognitive deficits.

## 2. Materials and Methods

### 2.1. Study Subjects and Sample Processing

The Newfoundland and Labrador Health Research Ethics Authority gave ethical approval for this study. Participants were recruited with invitation letters from the offices of family practitioners serving a high proportion of elderly clients. Exclusion criteria for the study included a diagnosis of dementia, previous history of negative reactions to blood sampling, acute infection, or known current immune disorder. Eligible individuals who responded were contacted by research staff, and if they expressed interest, an initial visit was scheduled. Informed consent for study participation was obtained during this visit, and psychological testing was carried out by trained personnel. Whole blood samples were collected into 10 mL acid citrate dextrose vacutainers by forearm venipuncture near the time of initial psychological testing. Plasma was collected following centrifugation of whole blood for 10 min at 500× *g*, and PBMC were isolated from the cellular fraction by Ficoll Hypaque (VWR Scientific, Mississauga, ON, Canada) density gradient separation. Plasma was immediately frozen at −80 °C and PBMC stored in liquid N_2_ in freeze medium made up of 90% fetal calf serum (FCS) from Invitrogen, Carlsbad, CA, USA, and 10% dimethyl sulfoxide (DMSO) from Sigma-Aldrich, St. Louis, MO, USA, until use.

### 2.2. Anti-CMV IgG ELISA

Antibodies against CMV were measured with a validated in house ELISA as previously described [23] Briefly, plasma was diluted 1:500 with phosphate-buffered saline (PBS) and incubated for 90 min in duplicate wells of Immulon-2 microtiter plates (VWR Scientific) coated with lysate from CMV-infected or uninfected MRC-5 cells (*ATCC*^®^ CCL-171™). Antibody binding was detected with goat-anti-human IgG-horseradish peroxidase conjugate (Jackson ImmunoResearch Labs, West Grove, PA, USA) followed by tetramethylbenzidine substrate (Sigma-Aldrich) and the difference in optical density (OD) at 405 nm between wells coated with CMV-infected versus uninfected MRC-5 cell lysate was measured on a Biotek synergy HT ELISA reader and recorded.

### 2.3. Measurement of Inflammatory Markers

Commercial ELISA kits were used to measure the following analytes in plasma as per the manufacturers’ instructions. Kits for measuring IL-1β (range = 2.00–200.00 pg/mL), IL-6 (range = 2.00–200.00 pg/mL), and tumor necrosis factor (TNF)-α (range = 4.00–500.00 pg/mL) were from eBioscience, San Diego, CA, USA. Kits for measuring fractalkine (range = 0.2–10 ng/mL) and C-reactive protein (CRP) (range = 15.60–1000.00 pg/mL) were from R&D Systems, Minneapolis, MN, USA. The sensitivity ranges spanned physiologically relevant levels such that plasma was added neat to the assay, except for the CRP assay where plasma was diluted 1:10,000 with PBS. All measurements were carried out in duplicate with control wells containing only PBS included in each assay. The OD value in control wells was subtracted from all test values to adjust for background. Absorbance was measured at 450 nm on a Biotek synergy HT ELISA reader and standard curves generated as specified by kit manufacturer’s instructions.

### 2.4. Measurement of CMV-Specific CD8^+^ T Cell Responses

Study participants were tested for CD8^+^ T cell responses against CMV pp65 and immediate early-1 (IE-1) proteins using sets of overlapping peptides (Peptivator, Miltenyi Biotec, San Diego, CA, USA). Aliquots of 2 × 10^6^ PBMC in 1 mL lymphocyte medium (RPMI 1640 with 10% FCS, 100 IU/mL penicillin, 100 µg/mL streptomycin, 2 mM l-glutamine, 10 mM HEPES buffer solution, and 2.0 × 10^−5^ M 2-mercaptoethanol from Invitrogen, Carlsbad, CA, USA) were stimulated with pooled CMV-pp65 (0.5 µg/mL) and IE-1 (0.5 µg/mL) peptide sets for 60 min at 37 °C (5% CO_2_). Brefeldin A (Sigma-Aldrich) was then added to a final concentration of 10.0 µg/mL and the PBMC left for an additional 4 h before staining for surface markers and intracellular interferon-gamma (IFN-γ). Cells were washed twice in flow cytometry buffer consisting of PBS with 0.2% NaN_3_, 5 mM ethylenediaminetetraacetic acid (EDTA, Sigma-Aldrich), and 0.5% FCS, then stained with fluorescein isothiocyanate-conjugated anti-human CD3, clone BW264/56, and peridinin chlorophyll protein-conjugated anti-human CD8, clone BW135/80, (Miltenyi Biotec), for 20 min at 4 °C. Samples were kept in the dark, and after another wash with flow buffer, cells were fixed and permeabilized with InsideStain (Miltenyi Biotec) according to manufacturer’s instructions, then stained with allophycocyanin-conjugated anti-human IFN-γ, clone 4S.83 (eBioscience).

### 2.5. Neuropsychological Testing

The Repeatable Battery for the Assessment of Neuropsychological Status (RBANS) was used to assess baseline cognitive functioning and possible cognitive decline in study participants [24]. The RBANS is an individually administered battery of tests that allows for the assessment of ability across five cognitive domains including immediate memory, delayed memory, visuospatial/constructional ability, language, and attention. The RBANS was administered at study entry and approximately 18 months afterwards. Appropriately trained individuals administered and scored the tests.

### 2.6. Mood

The Geriatric Depression Scale (GDS) was used to assess participants’ mood symptoms at baseline and 18 month follow-up. The GDS is a self-report measure consisting of 30 yes/no questions (https://doi.org/10.1300/J018v01n01_06 (accessed on 1 June 2021)).

### 2.7. Statistical Analysis

Statistical analyses were carried out using Prism (GraphPad Software, Inc., La Jolla, CA, USA). Normal distribution of data was assessed by the Shapiro–Wilk test, which showed deviation from normality in most cases. Therefore, data were represented with median ± interquartile range (IQR) and group medians compared by non-parametric Mann–Whitney tests. Correlation was used to assess relationships between variables using Spearman correlation matrices. For Mann–Whitney and Spearman correlation testing, *p*-values < 0.05 were considered significant. No correction for multiple testing was applied when calculating *p*-values.

## 3. Results

### 3.1. Study Subjects

Thirty-nine individuals (15 male, 24 female) between 70 and 90 years of age enrolled in this study (Table 1). A whole blood sample was drawn at time of enrollment, and plasma antibodies against CMV were measured to determine the CMV infection status of all subjects. Plasma levels of several cytokines and other markers of inflammation were also measured in this baseline sample. Testing for anti-CMV antibodies showed that 28 subjects were CMV-seropositive and 11 (5 males, 6 females) were CMV-seronegative (Figure 1A). For 36 subjects, including 25 of the 28 CMV-seropositive subjects, CD8^+^ T cell responses against the immunodominant CMV proteins pp65 and IE-1 were measured. Two of the 25 CMV-seropositive subjects tested, 1 male and 1 female, had CMV-specific CD8^+^ T cell responses involving >10% of their CD8^+^ T cell population (Figure 1B), a frequency indicative of CMV T cell immune memory inflation associate with an IRP. Standardized psychological testing was administered near the time of enrollment and approximately 18 months later. Five of the 39 subjects from whom blood samples were collected at study entry (three CMV-seropositive, two CMV-seronegative) were lost to follow-up before the second set of psychological tests could be carried out.

### 3.2. Plasma Markers of Inflammation

Plasma levels of the pro-inflammatory cytokines IL-1β, IL-6, and TNF-α together with the chemokine fractalkine (CX3CL1) and acute phase protein CRP were measured using commercial ELISA kits. A small fraction of subjects in both the CMV-seropositive and -seronegative groups had levels of these inflammatory markers above the normal range, but there were no significant differences between groups in median levels of any of the inflammatory markers (Figure 2). Thus, in this small group of elderly subjects, we saw no evidence of increased inflammation related to CMV infection or to CMV-specific CD8^+^ T cell memory inflation. There was a significant direct correlation between the magnitude of humoral and CD8^+^ T cell responses against CMV (*p* = 0.036), while the magnitude of CMV-specific CD8^+^ T cell responses negatively correlated with levels of both IL-1β (*p* = 0.031) and TNF-α (*p* = 0.049). When only CMV-seropositive individuals were considered, IL-1β levels directly correlated with TNF-α levels (*p* = 0.03), and IL-6 levels directly correlated with fractalkine levels (*p* = 0.001). The magnitude of the CMV-specific humoral immune response directly correlated with levels of fractalkine (*p* = 0.018), and the magnitude of CMV-specific CD8^+^ T cell responses negatively correlated with levels of TNF-α (*p* = 0.011). Levels of CRP did not correlate with any of the other pro-inflammatory markers. These correlations suggest that a number of independent and interdependent processes, some potentially related to the immune response against CMV, affect the inflammatory milieu in elderly individuals.

### 3.3. Psychological Testing

A set of standardized psychological tests, the RBANS, was administered by trained personnel to 39 subjects at or around study entry (t1) and to 34 subjects approximately 18 months after study entry (t2). Scores were collated and medians compared between groups of CMV-seropositive and CMV-seronegative subjects at both time points (Figure 3, Figure 4, Figure 5 and Figure 6). There were no significant differences detected between groups with scores from any of the neuropsychological tests at either time point. Changes observed over the 18 month follow-up period were generally minor and also similar between groups. As a relationship has been proposed between unhealthy aging and inflammation, we assessed correlations between levels of pro-inflammatory markers and scores obtained in the different categories of standardized testing. Age was inversely correlated with levels of IL-1β (*p* = 0.01) and TNF-α (*p* = 0.031), but directly correlated with fractalkine levels (*p* = 0.0039). At t1, TNF-α levels correlated directly with RBANS total (*p* = 0.031), immediate memory (ImMem) (*p* = 0.006), visuospatial/constructional (V/C) (*p* = 0.022), and delayed memory (DeMem) scores (*p* = 0.025). In contrast, fractalkine levels correlated inversely with ImMem scores (*p* = 0.01). At time 2, there were direct correlations of TNF-α levels with RBANS total (*p* = 0.026), language (*p* = 0.025), and DeMem scores (*p* = 0.02). Fractalkine levels correlated inversely with RBANS total (*p* = 0.0374), ImMem (*p* = 0.0145), and DeMem scores (*p* = 0.0196). When only CMV-seropositive subjects were considered, IL-1β levels correlated inversely (*p* = 0.043), and fractalkine levels correlated directly with age (*p* = 0.025). TNF-α levels correlated directly with RBANS total (*p* = 0.049) and ImMem scores (*p* = 0.005) at t1 and with RBANS total (*p* = 0.047) and DeMem scores (*p* = 0.029) at t2. Fractalkine levels correlated inversely with ImMem at t1 (*p* = 0.033) and t2 (*p* = 0.007) as well as with RBANS total (*p* = 0.022) and DeMem scores (*p* = 0.008) at t2. Fractalkine levels correlated directly with geriatric depression score (GDS) at t2 (*p* < 0.041). In the CMV-seropositive group of subjects, anti-CMV antibody levels correlated with GDS at t1 (*p* = 0.024) and t2 (*p* = 0.011). These data indicate a number of potential relationships between the immune response against CMV, the inflammatory milieu, and neuropsychological status of elderly persons (Figure 7A–D).

## 4. Discussion

In this study of elderly individuals (70–90 years old), we determined CMV serostatus and measured aspects of humoral and cellular immunity against CMV in those who were CMV-seropositive. To assess background levels of inflammaging, we also measured various plasma markers of inflammation. Since IL-6 is generally regarded as the major inflammatory driver of age-related pathology, we measured IL-6 along with IL-1β and TNF-α, which have a number of overlapping properties, including induction of the acute phase protein CRP, which was also measured [25]. Fractalkine was measured because of its relationship to neural excitotoxicity, microglial cell regulation, and because of the expression of a functional viral receptor for fractalkine on monocytes latently infected with CMV [26,27,28]. Cognitive ability and its decline were gauged with a series of standardized neuropsychological tests administered at study entry (t1) and repeated 18 months later (t2). 

While no significant differences were detected between groups of CMV-seropositive and CMV-seronegative subjects in median levels of markers of inflammation, cognitive test scores, or change in test scores over an 18 month period, we observed a number of significant correlations that warrant further investigation in more extensive studies. In terms of relationships between anti-CMV immunity and markers of inflammation, CMV-specific CD8^+^ T cell responses negatively correlated with levels of both IL-1β and TNF-α. Within the CMV-seropositive group, IL-1β levels directly correlated with TNF-α levels while IL-6 levels directly correlated with fractalkine levels. The magnitude of CMV-specific humoral immune responses also directly correlated with levels of fractalkine. Surprisingly, CRP levels did not correlate with any of the other pro-inflammatory markers. Interleukin-6 has previously been considered a marker of inflammaging [29,30], and in our groups of subjects, levels of the chemokine fractalkine were associated with levels of IL-6 and with levels of anti-CMV humoral immunity. Age inversely correlated with IL-1β and TNF-α levels, but directly correlated with fractalkine levels.

Discordant relationships between scores in the RBANS testing and different markers of inflammation also suggest that all inflammatory pathways are not equally related to cognitive decline, at least not in this small study group. At t1, TNF-α levels correlated directly with RBANS total, immediate memory (ImMem), visuospatial/constructional (V/C), and delayed memory (DeMem) scores. In contrast, fractalkine levels correlated inversely with ImMem scores. At time 2, TNF-α levels correlated directly with RBANS total, language and DeMem scores. Again, in direct contrast, fractalkine levels inversely correlated with RBANS total, ImMem, and DeMem scores. For CMV-seropositive subjects, fractalkine levels correlated directly with age and inversely with ImMem at t1 and t2 as well as with RBANS total and DeMem scores at t2. Fractalkine levels correlated directly with geriatric depression score (GDS) at t2, and in the CMV-seropositive group of subjects, anti-CMV antibody levels also correlated with GDS at t1 and t2.

Relationships suggested by data collected here will require corroboration in extended studies, but potential associations identified between fractalkine levels, anti-CMV humoral immunity, and age-related cognitive decline are of interest and consistent with a proposed role for fractalkine in regulating neuroinflammation. Fractalkine is highly expressed by neurons, and binding to its specific receptor CX3CR1 on glial cells acts to keep them in a quiescent, non-inflammatory state [28]. Monocytes and monocyte precursors latently infected with CMV express a CMV-encoded protein that mimics CX3CR1 and mediates chemotaxis of CMV-infected cells towards areas of higher fractalkine concentrations [27]. While preliminary, the interconnection of anti-CMV antibody levels (perceived as a surrogate marker for CMV reactivation frequency), fractalkine levels, and geriatric depression scores has implications in the context of a potential mechanistic link between depression and inflammation related to chronic CMV infection and recurrent reactivation of CMV throughout life. Relationships between anti-CMV antibody levels and cognitive deficits were previously reported in studies of middle-aged persons living with human immunodeficiency virus infection (PLWH) and in old elderly [20,21,31,32]. In the setting of HIV infection, the effects of CMV on the immune system are both accelerated and exaggerated, highlighting the relevance of CMV infection and anti-CMV immune status to the healthy aging of PLWH [33,34,35].

One of the difficulties with studies of age-related deterioration is the time required to effectively address questions posed. We saw little quantitative evidence of cognitive decline over 18 months in our study, which may suggest this is too short a time frame to pose these questions. In contrast to a previous study of over 500 subjects, we did not observe significant differences between CMV-seropositive and -seronegative groups with any of the individual RBANS tests or RBANS total [21]. Measuring global cognitive performance as a continuous measure and classifying cognitive impairment as a dichotomous construct could have reduced the number of comparisons. As this was an exploratory study, we did not correct for multiple testing, which increased the risk of falsely identifying some associations as significant. Some of the nominally significant findings require replication in larger studies. We selected inflammatory markers either previously associated with reduced cognition or with a mechanistic link to neuronal excitotoxicity for investigation [26,36,37]. Mechanisms underlying associations between chronic inflammation and reduced cognition remain speculative, but hippocampal volume loss and microglial activation have been proposed [38,39]. It remains plausible that inflammation affects cognition indirectly through a generalized negative effect on health and psychosocial functioning.

Other issues also limited the power to discriminate effects related to aging and the influence of anti-CMV immunity. As the study design involved recruitment of subjects responding independently to a written letter distributed at their doctor’s office, an inherent selection bias for highly educated subjects with stable cognitive properties may have been introduced. This also limited overall recruitment, and the loss of five persons to follow-up was significant. It may also have skewed towards recruitment of CMV-seronegative subjects as 11/39 subjects over the age of 75 years being CMV-seronegative is a higher proportion than expected. While meaningful data can be collected in smaller focused studies such as this one, ideally, they should be integrated within the framework of longitudinal studies of healthy aging where the numbers of participants and potential for long-term data collection is greater.

## 5. Conclusions

Within this group of elderly subjects, no common pattern of inflammaging related to anti-CMV immunity or age-related cognitive decline emerged, indicating that the relationship is complex. The apparent discordance between fractalkine levels and levels of other inflammatory markers, in terms of their relationship to anti-CMV immunity, geriatric depression scores, and cognitive issues in the elderly, may serve to focus aging studies on including the selective relevance of fractalkine levels in their analyses.

## Figures and Tables

**Figure 1 viruses-13-02321-f001:**
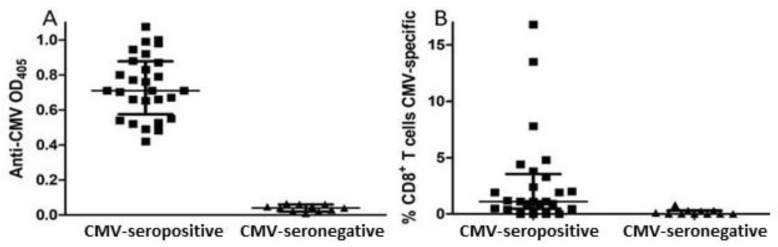
Humoral and CD8^+^ T cell responses against CMV in study participants. (**A**) The presence of IgG antibodies against CMV was assessed by semi-quantitative ELISA with plasma collected at time of entry. (**B**) PBMC were tested for CD8^+^ T cell responses against peptide pools spanning CMV pp65 and IE-1 proteins by intracellular flow cytometry detecting IFN-γ. Error bars on graphs show median ± interquartile range for each group.

**Figure 2 viruses-13-02321-f002:**
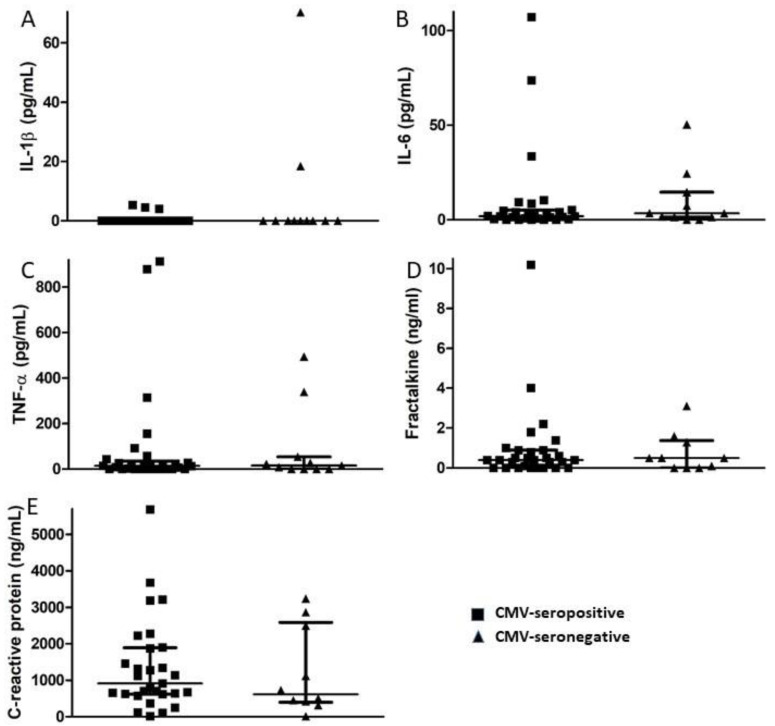
Comparison of plasma levels of inflammatory markers in CMV-seropositive and CMV-seronegative study participants. Plasma levels of the pro-inflammatory cytokines (**A**) IL-1β, (**B**) IL-6, (**C**) TNF-α, (**D**) the chemokine fractalkine, and (**E**) acute phase protein CRP were measured by ELISA at the time of study entry for each participant and median values compared between CMV-seropositive and CMV-seronegative groups. Error bars on graphs show median ± interquartile range for each group. No significant differences between CMV seropositive and –seronegative groups were observed. Mann–Whitney *U*-test, (**A**) *p* = 0.452, (**B**) *p* = 0.584, (**C**) *p* = 0.939, (**D**) *p* = 0.909, (**E**) *p* = 0.531.

**Figure 3 viruses-13-02321-f003:**
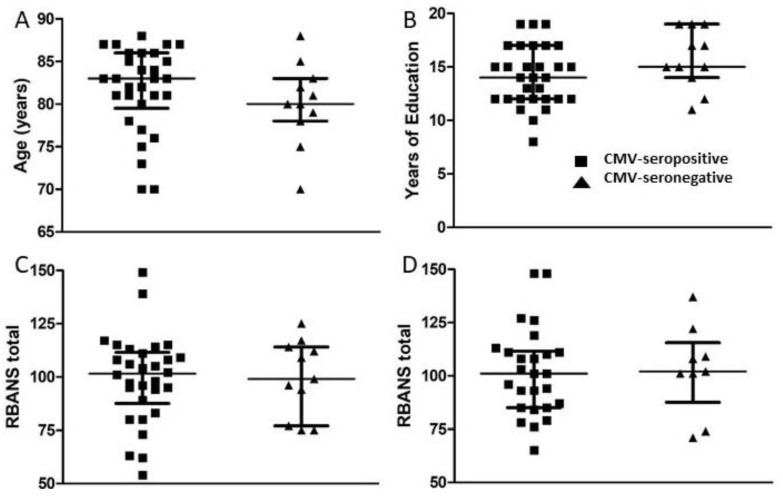
Comparison of age, years of education, and RBANS total scores between groups of CMV-seropositive and CMV-seronegative subjects. (**A**) Age and (**B**) years of education are represented as at study entry (t1) for all subjects. Following neuropsychological testing, RBANS total scores were calculated at t1 for all study subjects (**C**) and approximately 18 months later (t2) for all study subjects completing follow-up (**D**). Error bars on graphs show median ± interquartile range for each group. No significant differences between CMV-seropositive and -seronegative groups were observed. Mann–Whitney *U*-test, (**A**) *p* = 0.215, (**B**) *p* = 0.128, (**C**) *p* = 0.883, (**D**) *p* = 0.895.

**Figure 4 viruses-13-02321-f004:**
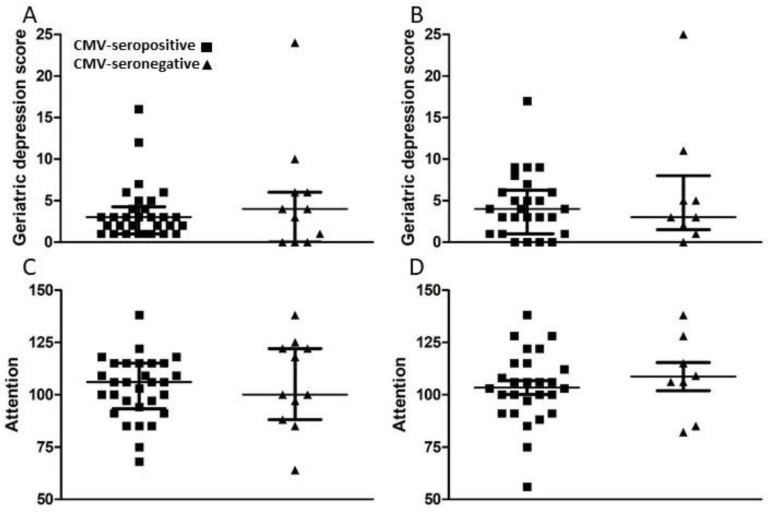
Comparison of geriatric depression and attention scores between groups of CMV-seropositive and -seronegative subjects at t1 and t2. (**A**,**C**) Following neuropsychological testing, geriatric depression and attention scores were calculated at t1 for all study subjects and (**B**,**D**) approximately 18 months later (t2) for all study subjects completing follow-up. Error bars on graphs show median ± interquartile range for each group. No significant differences between CMV-seropositive and -seronegative groups were observed. Mann–Whitney *U*-test, (**A**) *p* = 0.766, (**B**) *p* = 1.000, (**C**) *p* = 0.605, (**D**) *p* = 0.442.

**Figure 5 viruses-13-02321-f005:**
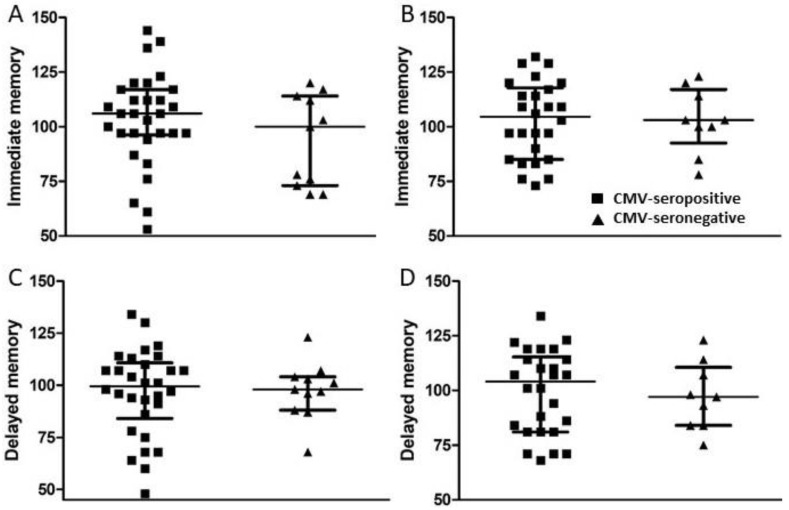
Comparison of immediate and delayed memory scores between groups of CMV-seropositive and -seronegative subjects at t1 and t2. (**A**,**C**) Following neuropsychological testing, immediate and delayed memory scores were calculated at t1 for all study subjects and (**B**,**D**) approximately 18 months later (t2) for all study subjects completing follow-up. Error bars on graphs show median ± interquartile range for each group. No significant differences between CMV-seropositive and -seronegative groups were observed. Mann–Whitney *U*-test, (**A**) *p* = 0.353, (**B**) *p* = 0.940, (**C**) *p* = 0.871, (**D**) *p* = 0.777.

**Figure 6 viruses-13-02321-f006:**
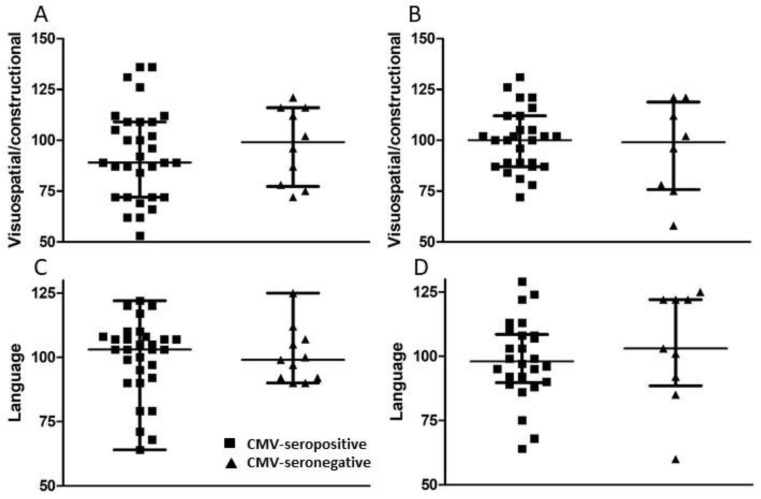
Comparison of visuospatial/constructional and language scores between groups of CMV-seropositive and -seronegative subjects at t1 and t2. (**A**,**C**) Following neuropsychological testing, visuospatial/constructional and language scores were calculated at t1 for all study subjects and (**B**,**D**) approximately 18 months later (t2) for all study subjects completing follow-up. Error bars on graphs show median ± interquartile range for each group. No significant differences between CMV-seropositive and -seronegative groups were observed. Mann–Whitney *U*-test, (**A**) *p* = 0.429, (**B**) *p* = 0.708, (**C**) *p* = 0.701, (**D**) *p* = 0.450.

**Figure 7 viruses-13-02321-f007:**
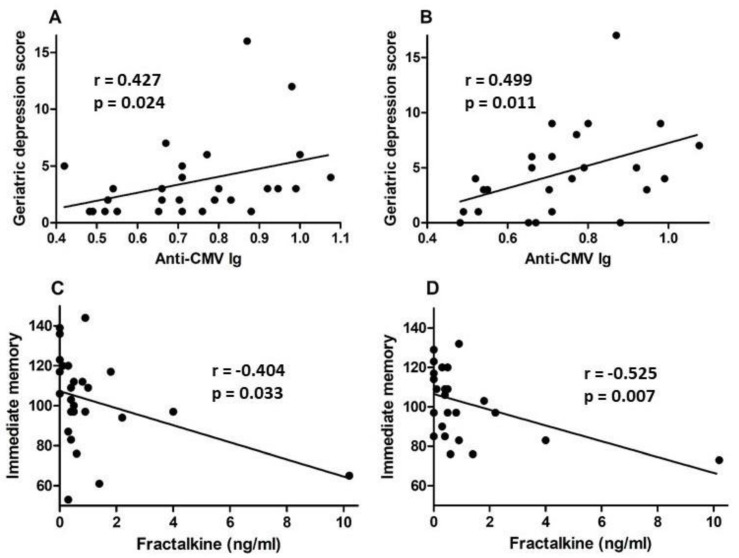
Correlation of anti-CMV IgG and fractalkine levels with neuropsychological test scores for CMV-seropositive subjects. Anti-CMV IgG levels in plasma were plotted against geriatric depression scores from neuropsychological testing at (**A**) study entry and (**B**) approximately 18 months later. Plasma fractalkine levels were plotted against immediate memory scores from neuropsychological testing at (**C**) study entry and (**D**) approximately 18 months later. Correlation was assessed by linear regression and correlation coefficient (r) and significance values (*p*) shown within each plot frame.

**Table 1 viruses-13-02321-t001:** Study participant general characteristics.

Participant ID	Age at Entry	Years of Education	Sex	CMV Status
3001	83	12	Female	Seronegative
3002	87	10	Female	Seropositive
3003	84	12	Male	Seropositive
3004	83	14	Female	Seropositive
3005	81	15	Male	Seronegative
3006	88	12	Male	Seropositive
3007	80	17	Female	Seropositive
3008	81	15	Male	Seropositive
3009	83	12	Female	Seropositive
3010	87	12	Female	Seropositive
3011 *	82	11	Female	Seronegative
3012	87	8	Female	Seropositive
3013	85	14	Female	Seropositive
3014	81	19	Male	Seropositive
3015	84	15	Female	Seropositive
3016 *	83	11	Male	Seropositive
3017	85	19	Male	Seropositive
3018 *	88	15	Male	Seronegative
3019	80	15	Female	Seronegative
3020 *	87	14	Female	Seropositive
3021	86	17	Female	Seropositive
3022	85	14	Female	Seronegative
3023	86	12	Female	Seropositive
3024	82	15	Male	Seropositive
3025 *	86	11	Female	Seropositive
3026	83	15	Male	Seropositive
3027	78	19	Male	Seropositive
3028	75	17	Female	Seronegative
3029	70	17	Female	Seronegative
3030	78	19	Male	Seronegative
3031	80	19	Male	Seronegative
3032	82	17	Female	Seropositive
3033	79	19	Male	Seronegative
3034	73	15	Female	Seropositive
3035	76	17	Male	Seropositive
3036	75	13	Female	Seropositive
3037	81	12	Female	Seropositive
3038	70	12	Female	Seropositive
3039	70	17	Female	Seropositive

* Did not complete 18 month follow-up neuropsychological testing.

## Data Availability

The datasets used and/or analysed during the current study are available from the corresponding author on reasonable request.

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
