# Peer review of "Cytomegalovirus Immunity, Inflammation and Cognitive Abilities in the Elderly"

_viruses, 2021, doi:10.3390/v13112321_

Round 1

Reviewer 1 Report

This article titles Cytomegalovirus Immunity, Inflammation and Congnitive Abilities in the Elderly by Hesson et al looks at possible links between CMV infection, immune senescences and cognitive impairment in the elderly. I found this article to be an interested area to address. However, I did have several questions on the limitation of their methods. 

  1. Why was only 1 dilution of 1:500 tested for anti-CMV ELISA? Do you not miss lower positive results as in if you tested at 1:40 or 1:80 dilution?
  2. In measuring cytokine activity, why was IL-8 not on the list? 
  3. Are there any links to CMV titer and cytokine or CD8 T cell responses? There seem to be high or low outliers inflammation markers and memory tests. Are there any links to CMV titer?
  4. I felt that overall, figures do not show any differences in most of the assays between CMV+ and CMV- samples. However, there are positive and negative correlations made in the text. It would have been more helpful is these can be graphically presented.

Author Response

Thank you for your positive comments and for raising relevant questions.

  1. Why was only 1 dilution of 1:500 tested for anti-CMV ELISA? Do you not miss lower positive results as in if you tested at 1:40 or 1:80 dilution?

In a previous study, referenced in the methods section, we found that a 1:500 dilution of plasma gave the best results in terms of low background binding to the uninfected cell lysate and a broad dynamic range across the level of responses in the CMV-seropositive population. As shown in Figure 1A, this provides excellent discrimination between CMV-seropositive and CMV-seronegative subjects without ambiguous results.

  1. In measuring cytokine activity, why was IL-8 not on the list? 

We chose to measure cytokines (IL-1, IL-6, TNF) and other markers (C-reactive protein) that are considered indicative of chronic low grade chronic inflammation that occurs with aging unrelated to acute infection or viral reactivation. While IL-8 would likely be elevated in acute CMV infection or with CMV reactivation, our purpose in measuring inflammatory markers was not to detect active infection or assess relationships with active infection but to assess low-grade chronic inflammation associated with aging.

  1. Are there any links to CMV titer and cytokine or CD8 T cell responses? There seem to be high or low outliers inflammation markers and memory tests. Are there any links to CMV titer?

None of the subjects showed signs of active CMV infection at the times they were tested and we did not measure CMV viral titer. There was a significant positive correlation between anti-CMV antibody levels and CD8+ T cell responses.  Higher antibody levels have been used to indicate greater lifelong exposure to CMV reactivation, so the positive correlations supports the idea that the higher CD8+ T cell responses are related to more CMV reactivation.

  1. I felt that overall, figures do not show any differences in most of the assays between CMV+ and CMV- samples. However, there are positive and negative correlations made in the text. It would have been more helpful is these can be graphically presented.

It is true that in this small study, no significant differences markers of inflammation or neuropsychological status were observed between the CMV-seropositive and –seronegative groups. Several of the potentially more relevant significant correlations are now graphically illustrated in an additional figure (figure 7).  

Reviewer 2 Report

The authors compare elderly but healthy volunteers to find out if CMV-immune status (primarily antibodies and in Fig. 2 also CD8+T cells) has an impact of various parameters tested, including parameters of inflammation and parameters of cognitive ability. They also compare CMV-seropositive and CMV-seronegative subjects in an 18-months follow-up. In essence, significant differences were not found in any of these comparisons.

Apparently, the authors are not experts for CMV. This explains why the rationale for the study is poor. In the Introduction, CMV-related references are highly selected, mostly not up-to-date, and not representative of the current state of discussion in this field. More recently, even previous protagonists of the hypothesis of CMV-associated "immunosenescence" related to the phenomenon of "memory inflation" have dampened or even revised their views and a biennial workshop to this issue now avoids to use "immunosenescence" in its title. Thus, overall, the referencing of the paper is inappropriate and does not reflect the current state of this field. This narrowed view affects the interpretation and weakens the discussion of the data. 

One aspect indicative of a misinterpretation by the authors is the use of the term "chronic infection". Apparently, the authors are unaware of the phenomenon of "latent infection/latency" that applies to CMV. An impact of CMV infection status on inflammatory markers is not to be expected during viral latency but likely would have been found during episodes of recurrent infection. Unfortunately, the infection status (not just the immune status) was not tested, although reactivated/recurrent infection might have explained some of the high values seen in Figure 2. Likewise, changes within the 18-months follow-up could be expected only for rare cases of virus reactivation/recurrence occurring in this narrow period (not narrow for investigation, but nevertheless narrow in relation to life-time latent infection).

Minor point: statistics is documented with error bars, but P-values would reveal more clearly that the differences are not-significant throughout.       

Author Response

Apparently, the authors are not experts for CMV. This explains why the rationale for the study is poor. In the Introduction, CMV-related references are highly selected, mostly not up-to-date, and not representative of the current state of discussion in this field. More recently, even previous protagonists of the hypothesis of CMV-associated "immunosenescence" related to the phenomenon of "memory inflation" have dampened or even revised their views and a biennial workshop to this issue now avoids to use "immunosenescence" in its title. Thus, overall, the referencing of the paper is inappropriate and does not reflect the current state of this field. This narrowed view affects the interpretation and weakens the discussion of the data. 

Thank you for your comments and critique on our manuscript. While not claiming to be CMV experts, we are aware of controversy around the proposed link between CMV infection and immunosenescence. We have commented in previous publications that using the term immunosenescence in reference to accumulation of terminally differentiated memory cells and some level of immune dysfunction is inappropriate (Heath et al. Cells 2020, 9, 766; doi:10.3390/cells9030766). Our working hypothesis is that in the more extreme cases of memory inflation, the CMV responsive immune cells are approaching replicative senescence at which point they would acquire the senescence-associated secretory phenotype and contribute to low grade systemic inflammation. We have shown by measuring telomere length and T cell receptor excision circle frequency in persons living with HIV that CMV-specific CD8+ T cells are the lymphocytes closest to replicative senescence (Barrett et. al. Clin. Infect. Dis. 2016, 62:1467-8, Heath et al. Front. Immunol.2018 doi.org/10.3389/fimmu.2018.00201). We believe that the same is true of CMV-specific CD8+ T cells in elderly persons with CMV memory inflation.  Literature cited in the introduction supports this working hypothesis and our discussion is in part shaped around it. As far as we know, none of the more recent literature refutes the fact that the immune response to CMV often exhibits memory inflation, producing an expanded pool of memory CD8+ T cells that have undergone extensive replication. We agree that the term immunosenescence should not be used to describe this phenomenon and have attempted to clarify our position around this topic with additional text in the introduction.  

One aspect indicative of a misinterpretation by the authors is the use of the term "chronic infection". Apparently, the authors are unaware of the phenomenon of "latent infection/latency" that applies to CMV. An impact of CMV infection status on inflammatory markers is not to be expected during viral latency but likely would have been found during episodes of recurrent infection. Unfortunately, the infection status (not just the immune status) was not tested, although reactivated/recurrent infection might have explained some of the high values seen in Figure 2. Likewise, changes within the 18-months follow-up could be expected only for rare cases of virus reactivation/recurrence occurring in this narrow period (not narrow for investigation, but nevertheless narrow in relation to life-time latent infection).

To our understanding, CMV infection, as with other herpes viruses, is always termed a chronic infection, whether there is ongoing replication or the virus is only persisting as DNA in particular cells. Latent CMV infection is not necessarily silent as gene products expressed during latency include CMV IL-10 and fractalkine receptor homologues. We were not attempting to relate the inflammatory markers (IL-1, IL-6, TNF and CRP) to active CMV replication, but wanted to test the possibility that that the long term effects of CMV on the immune system might affect low grade inflammation associated with unhealthy aging. Admittedly, we did not see changes in measures of cognition in community dwelling, healthy elderly participants over an 18 month period, suggesting that longer term follow-up would be required to observe changes and test for any relationship with CMV immunity.  This limitation is mentioned in the discussion.  We have not addressed the question of whether CMV reactivation itself could play a role in chronic low grade inflammation or cognitive decline.  Other text has been added to clarify our focus was not on  direct effects of CMV reactivation.

Minor point: statistics is documented with error bars, but P-values would reveal more clearly that the differences are not-significant throughout.   

P-values have now been included in the captions of relevant figures.

Reviewer 3 Report

An interesting paper that reads well and has a good standard of presentation.  The title is relevant.  The introduction is comprehensive, balanced, and informative.  The methods are described in adequate detail.  The results are presented to an adequate standard.  The discussion is balanced, relevant, and reviews the study limitations to a satisfactory standard. 

Author Response

Thank you for your comments on our manuscript.

Round 2

Reviewer 2 Report

Change title to give the message that CMV infection status has no impact on the parameters tested. The current title is too general and fails to reflect the content and message of the paper.

The referencing ist still inadequate.